# Bacterial Colonization within the First Six Weeks of Life and Pulmonary Outcome in Preterm Infants <1000 g

**DOI:** 10.3390/jcm9072240

**Published:** 2020-07-15

**Authors:** Tina Lauer, Judith Behnke, Frank Oehmke, Johanna Baecker, Katrin Gentil, Trinad Chakraborty, Michael Schloter, Jan Gertheiss, Harald Ehrhardt

**Affiliations:** 1Department of General Pediatrics and Neonatology, Justus-Liebig-University, Universities of Gießen and Marburg Lung Center (UGMLC), German Center for Lung Research (DZL), Feulgenstrasse 12, D-35392 Gießen, Germany; tina.lauer@paediat.med.uni-giessen.de (T.L.); judith.behnke@paediat.med.uni-giessen.de (J.B.); johanna.baecker@paediat.med.uni-giessen.de (J.B.); 2Department of Obstetrics and Gynecology, Justus-Liebig-University, Feulgenstrasse 12, D-35392 Gießen, Germany; Frank.Oehmke@gyn.med.uni-giessen.de; 3Institute of Medical Microbiology, Justus-Liebig-University, German Center for Infection Research (DZIF), Schubertstrasse 81, D-35392 Gießen, Germany; katrin.gentil@mikrobio.med.uni-giessen.de (K.G.); Trinad.Chakraborty@mikrobio.med.uni-giessen.de (T.C.); 4Research Unit for Comparative Microbiome Analysis, Helmholtz Zentrum München GmbH, Ingolstädter Landstrasse 1, D-85764 Neuherberg, Germany; schloter@helmholtz-muenchen.de; 5Department of Economics and Social Sciences, Statistics and Data Science Group, Helmut Schmidt University, Holstenhofweg 85, D-22043 Hamburg, Germany; jan.gertheiss@hsu-hh.de

**Keywords:** preterm infant, bronchopulmonary dysplasia, bacterial colonization, pathogenic, gram-negative bacteria, upper airway, lung, gut

## Abstract

Bronchopulmonary dysplasia (BPD) is a multifactorial disease mainly provoked by pre- and postnatal infections, mechanical ventilation, and oxygen toxicity. In severely affected premature infants requiring mechanical ventilation, association of bacterial colonization of the lung and BPD was recently disclosed. To analyze the impact of bacterial colonization of the upper airway and gastrointestinal tract on moderate/severe BPD, we retrospectively analyzed nasopharyngeal and anal swabs taken weekly during the first 6 weeks of life at a single center in *n* = 102 preterm infants <1000 g. Colonization mostly occurred between weeks 2 and 6 and displayed a high diversity requiring categorization. Analyses of deviance considering all relevant confounders revealed statistical significance solely for upper airway colonization with bacteria with pathogenic potential and moderate/severe BPD (*p* = 0.0043) while no link could be established to the Gram response or the gastrointestinal tract. Our data highlight that specific colonization of the upper airway poses a risk to the immature lung. These data are not surprising taking into account the tremendous impact of microbial axes on health and disease across ages. We suggest that studies on upper airway colonization using predefined categories represent a feasible approach to investigate the impact on the pulmonary outcome in ventilated and non-ventilated preterm infants.

## 1. Introduction

Bronchopulmonary dysplasia (BPD) is a chronic lung disease of preterm infants leading to life-long restrictions in lung function even in infants who do not fulfill the clinical criteria of BPD of dependency on mechanical ventilation or oxygen at a corrected age of 36 weeks of gestation [1,2]. BPD evokes from an inflammatory response in the immature lung that is dominated by pro-inflammatory cytokines and the influx of inflammatory cells causing dysregulation of the highly complex program of further lung development. Mechanical ventilation and oxygen toxicity constitute well accepted key drivers of the imbalance of pro-inflammatory properties and anti-inflammatory and lung growth promoting signaling pathways. The multifactorial origins of BPD include genetic predisposition, intrauterine growth restriction, surfactant deficiency, postnatal fluid management, and nutritional supply [3,4]. The impact of pre- and postnatal infections on lung maturation was not established until recently. Meanwhile, meta-analyses clearly demonstrate the overall negative impact of infections on lung development [5,6]. Further in-depth analysis is urgently required. Bacterial colonization can also exert beneficial features inducing immunotolerance and modifying the corticosteroid stress response [7,8,9,10].

For a long time, the amniotic cavity has been considered a sterile environment and any bacterial penetration an agitator of inflammation in the fetus and preterm birth. In fact, physiologically the fetus is exposed to a changing microbiome in utero with reduced bacterial richness and diversity and increased bacterial biomass over time [11,12,13,14]. Recently, microbial dysbiosis was observed in the lung of severely affected preterm infants developing BPD after delivery and during the course of mechanical ventilation. Preexisting microbial dysbiosis at birth and rapid postnatal shifts in bacterial composition were identified as key features of the bacterial dysbiosis. Microbial dysbiosis was recently described to be associated with changes in the lung metabolome and BPD emphasizing its clinical relevance [15,16,17]. Further evidence for the importance of the microbial dysbiosis has been derived from an observational study of preterm survivors that displayed alterations in the airway microbiome even in adults, who survived BPD as preterm infants [18]. Overall, there is strong evidence that bacterial dysbiosis of the lung strongly impacts the immature lung comparable to the well acknowledged effect of shifts in the gut microbiome of preterm infants and necrotizing enterocolitis (NEC) [19,20]. These similarities are not surprising as the lung and gut both develop from the foregut which separates in early embryonic evolution and display many common features.

The clinical measures of breast milk supply and reduction of antibiotic exposure are well known to have a positive impact on the gut microbiome development and to reduce the incidence of NEC. Breast milk supply increases microbial diversity and enhances immunologic properties while antibiotic therapy is a major cause of microbial dysbiosis. In the context of lung development and lung injury, meta-analyses of breast milk supply display a relevant reduction in the incidence of BPD [21,22]. Furthermore, recent studies demonstrated that antibiotic therapy per se and its prolonged use was associated with an increased risk for BPD [23,24,25]. Similarly, late onset sepsis increases the risk for BPD [26,27]. These consistent results from observational studies in preterm infants are not surprising taking into account the overall tremendous impact of shifts in microbial diversity and function of the gut and lung associated with acute and chronic diseases across all ages [10]. For the gut, the benefits of therapeutic interventions with probiotic supplementation have been clearly demonstrated. A similar effect on BPD could not be demonstrated but was not the primary endpoint of the studies conducted [28,29].

The high throughput analysis of microbiomes by molecular barcoding is currently restricted to research projects and not standard of clinical routine. Due to these limitations, we used a classical culture-based approach to assess bacterial diversity of the upper airway and gastrointestinal tract on a weekly basis to identify preterm infants with a birth weight <1000 g and gestational age ≤32 + 0 weeks at special risk for BPD. The inclusion criteria were selected based on the particularly high vulnerability for BPD in this population of infants with lungs in the saccular stage of lung development [30,31]. The screening approach to the nasopharynx and gut was devised to cover all infants at risk and receiving all forms of respiratory support, as more and more preterm infants can be stabilized by non-invasive mechanical ventilation or do not require prolonged invasive ventilation. We evaluated in detail the bacterial diversity pattern within the first six weeks of life which constitute the decisive period from birth where most infants were colonized.

## 2. Experimental Section

### 2.1. Patient Cohort, Exclusion Criteria, and Ethics Vote

Preterm infants with a birth weight <1000 g and gestational age ≤32 + 0 weeks were analyzed within a retrospective cohort study at our tertiary perinatal center (Justus-Liebig-University Giessen, Germany) between January 2014 and June 2017. Overall, *n* = 146 inborn patients were eligible. *n* = 37 preterm infants were excluded due to death before 36 weeks of gestation (*n* = 28), severe congenital malformations of the heart, brain, gut, or urogenital tract (*n* = 7) or transfer to another center before 36 weeks of gestation (*n* = 2). Furthermore, all infants with severe gastrointestinal complications (*n* = 7) of necrotizing enterocolitis or focal intestinal perforation were excluded due to the high impact of prolonged ventilation periods and inflammatory episodes on the pulmonary outcome. Overall, *n* = 102 infants were included in the initial analyses. Two infants had to be excluded from further analyses due to missing information on more specific bacterial colonization.

The study was conducted following the rules of the Declaration of Helsinki of 1975, revised in 2013. The retrospective analysis was approved by the ethics committee of the Justus-Liebig-University Gießen (Az 97/14).

### 2.2. Longitudinal Screening for Bacterial Colonization

According to the recommendations by the German Commission for Hospital Hygiene and Infectious Disease Prevention at the Robert Koch Institute (RKI) in 2013, bacterial colonization of the upper airway, and the gastrointestinal tract of all preterm infants were screened by nasopharyngeal and anal swabs, respectively, during routine clinical care at birth and weekly thereafter. Nasopharyngeal and anal swabs (Microbiotech S.r.l., Maglie, Italy) were taken during daily routine by trained ward staff. Samples were promptly transferred to the microbiological laboratory. Swabs were routinely cultured on blood, chocolate, and MacConkey agar plates (Thermo Fisher Scientific, Waltham, MA, USA). Plates were incubated at 37° with 5% carbon dioxide. Cultures were analyzed after 16–22 h and again after 48 h of incubation. Bacteria were analyzed by MALDI-TOF technology (VITEK MS, bioMérieux Deutschland GmbH, Nurtingen, Germany). For our analysis, bacterial strains were categorized according to Gram staining. Additionally, twelve species were classified as potential pathogens based on published literature and datasets from the German National Reference Center for the Surveillance of Nosocomial Infections (summarized in Table 1) [32,33,34,35]. Due to the high prevalence of colonization with coagulase-negative staphylococci in 95 out of 102 infants (93.1%), a statistical analysis to dissect its impact on BPD was not applicable to our cohort.

### 2.3. Data Acquisition and Parameter Definition

All metadata from patients were extracted from the electronic data management systems and the archived paper file records. Data were entered into a SPSS Statistics databank version 26.0.0.0 (IBM, Armonk, NY, USA) for Microsoft Windows (Redmond, WA, USA). The following baseline maternal and infant parameters as well as disease stages were recorded: gestational age, birth weight, gender, birth as singleton or multiple, antenatal steroid application, intraventricular hemorrhage, periventricular leukomalacia, necrotizing enterocolitis, and retinopathy of prematurity. Antenatal steroids were recorded as days before delivery or categorized as no completed course including four cases with no antenatal steroid application before birth (<24 h), delivery within 24 h and 7 days after the last application and provision of antenatal steroids >7 days before birth. Small for gestational age (SGA) status was defined as all three parameters of birth weight, length, and head circumference below the 10th percentile according to the percentiles from the German perinatal registry [36]. Severity of BPD was separated according to the current NIH consensus definition as no, mild, moderate or severe BPD [2]. To assess the level of Continuous Positive Airway Pressure (CPAP) under highflow nasal cannula support and the fraction of oxygen provided by nasal cannula, the published conversions were applied as before. A positive end-expiratory pressure of ≥3 cm H_2_O was estimated as CPAP [37,38]. Provision of breast milk from the baby’s mother was supported in the absence of a milk donor program and breast milk was fortified with Aptamil FMS 4,4% (Milupa, Frankfurt am Main, Germany). Nutritional supply was provided according to the actual recommendations and parenteral nutrition was discontinued when nutritional requirements were met by the enteral supply [39]. Furthermore, antibiotic exposure and the total days of antibiotic therapy within the first six weeks of life were recorded.

### 2.4. Statistical Analysis

To dissect the impact of specific patterns of microbial colonization and BPD, we focused our analysis on preterm infants with a birth weight <1000 g and below 32 weeks of gestation that have the highest risks for relevant limitations in lung function (*n* = 102) [31]. Baseline cohort characteristics were compared using a Mann–Whitney U test and SigmaPlot Version 12.3 (Systat Software Inc., San Jose, CA, USA). As all infants <1000 g display some restriction in lung function and most of them fulfill the BPD criterion, statistical analysis was directed towards the separation of BPD disease severities by comparing infants without/with mild BPD and infants with moderate/severe BPD. Comparisons were done using logistic regression in R 2.5.2 (R Base Distribution, R Core Team, Vienna, Austria). All datasets are presented after risk adjustment for gestational age, birth weight, small for gestational age (SGA), gender, multiple birth, provision of antenatal steroids, and antibiotic exposure. As indicated, different definitions of antibiotic exposure and provision of breast milk were analyzed as well. The corresponding confounder model was chosen via AIC-based, backward selection; with known risk factors gestational age and multiple births added manually. Effects of bacterial colonization were investigated by Analysis of Deviance. Statistical significance was accepted at *p* < 0.05.

## 3. Results

In our cohort, median gestational age was 26 + 3 (range 23 + 4 to 30 + 5) and median birth weight 795 g (range 320 to 995 g). As expected, most infants of the total cohort fulfilled any severity stage of BPD (82.4%) and about one third (34.3%) the criterion of moderate/severe BPD (further details are depicted in Table 2). As all former very premature infants display restrictions in lung function later in life, infants were separated according to the extent of functional limitations into no/mild (*n* = 67) and moderate/severe (*n* = 35) BPD.

### 3.1. Bacterial Colonization of the Upper Respiratory Tract and the Gut within the First Six Weeks of Life

Due to the improvements in non-invasive ventilation strategies during the last few decades, 42 of 102 (41.2%) infants from the total cohort never required invasive mechanical ventilation excluding tracheal aspirates as applicable microbiological source. Instead, swabs from the upper airway were analyzed. We suggest that the upper respiratory tract is at least partially shaped by the bacterial milieu of the lung via mucociliary clearance. The gut–lung axis has been described as another important contributor to pulmonary health and disease in pediatric and adult patients [10]. Therefore, anal swabs as surrogate parameter for gut colonization were included. Upper airway and gut bacterial colonization within the first six weeks of life displayed a high heterogeneity within the total cohort. Frequencies of specific strains were within the expected range, but variations in individual species and multiresistant bacteria for each detection site were insufficient for statistical analyses despite the total detection rates of up to 70.6% when all sites and time points were added.

Consideration of all bacterial strains for both localizations revealed low frequencies at birth but rapid colonization with frequencies approximating 80% in week 2 excluding a scientific approach (Figure 1A,B). Therefore, germ classification was applied. Bacterial strains were separated first according to their Gram staining properties or second into bacteria with and without pathogenic potential to the preterm infant (Table 1) [32,33,34,35]. Immediately after birth, bacteria were detected rarely for both bacterial categories. Commencing in week 2, the number of infants with upper airway and gut colonization steadily increased allowing statistical analyses (Figure 2A–D). To avoid statistical/numerical problems associated with very low incidences in week 1, weeks 1 and 2 were combined as “≤2” for further analyses.

After the first six weeks of life, most infants were colonized (Figure 2A–D). Due to the limited sample size, the distributions of first bacterial detection displayed relatively low incidences (absolute frequencies) for nearly each week, preventing a more detailed statistical approach on a weekly basis.

### 3.2. Upper Airway Colonization with Potential Pathogenic Bacteria Separates BPD Disease Severities

Similar to the published literature, infants in the moderate/severe BPD group were more immature, had a lower birth weight and displayed a higher frequency of male gender and small for gestational age status. Antenatal steroid exposure was not equally distributed between the groups (Table 2). To compensate for relevant confounders in further analyses, these variables plus the variables multiple birth and duration of antibiotic therapy within the first six weeks of life were included into the risk adjustment as obtained via AIC-based backward selection in a logistic regression model fitted via maximum likelihood (Table 3) [40].

To elaborate a significant impact of each bacterial category on the severity of BPD, the confounder model was compared to the model with the respective germ category (additionally) included using analysis of deviance. Within all studied variations of upper airway and gut bacterial colonization, only the colonization of the upper respiratory tract with strains with pathogenic potential prevailed statistically significant (*p* = 0.0043, Table 4). Gram-negative bacteria and anal swabs did not even display a trend towards statistical significance. Statistical analysis was repeated after excluding infants from the category “no detection of bacteria until week 6”. However, results did not change substantially (*p* = 0.0038). Table 5 gives the results for the final (logit) model regressing BPD (moderate/severe vs. no/mild) on oral pathogenic bacterial colonization and additional, potentially confounding covariates (as described above). Information on bacterial colonization (i.e., week) is dummy-coded with the first category “week ≤2” chosen as the reference; this means that estimated regression coefficients for bacterial colonization give changes in the log odds when compared to the reference. As displayed in the description of colonization, systematic evaluation of the time course of colonization with specific germs was not feasible due to the low frequencies and variations between the groups. However, (negative) coefficients of upper airway pathogenic bacterial colonization in Table 5 indicate a drop in the risk of BPD after week 2, even though the risk was not monotonically decreasing across weeks for the data at hand. Consequently, it may only be stated at this point that the, albeit statistically significant, effect for bacteria with pathogenic potential in the upper airway was unstructured and not directly attributable towards a specific time point. This observation might be due to opposing effects by different bacterial species within the category. Linear modeling of weeks for the datasets of total oral bacterial colonization (results not shown in detail here), however, also revealed a significant overall impact (*p* = 0.0164) and a risk reduction from week 3 on (*p* = 0.0204) pointing towards an overall unfavorable impact of early occurrence.

When fitting and comparing the considered models, it is assumed that observations are independent given the covariates. This assumption, however, might be questionable for multiples included in the data. For comparison, we hence fitted a generalized mixed model with family-specific random intercept as well. The estimated variance component, however, turned out to be (numerically) zero, leading to the equivalent models and same test results as described above (data not shown).

### 3.3. No Significant Impact of Varied Definition of Antibiotic Exposure and Breast Milk Supply

To further dissect the impact of antibiotic exposure on the colonization of the upper airway with bacteria with pathogenic potential in the study cohort, the total duration of antibiotic therapy measured in days was replaced by the covariates “antibiotic therapy started immediately after birth” or “any antibiotic therapy within the first 6 weeks of life”. Corresponding modeling did not result in relevant changes in *p*-values for the other covariates analyzed (Table 6 and Table 7). The statistical results for the colonization of the upper airway with bacteria with pathogenic potential and BPD severity separator were not altered substantially (Table 8). In accordance, Kendall’s tau (0.2473) did not reveal substantial correlation between the number of days antibiotics were given and the week bacteria with pathogenic potential were found in the upper respiratory tract for the first time.

As the provision of breast milk has a tremendous well recognized impact on the microbiota structures in the preterm infant, the impact of the addition of this covariate was analyzed separately. Extension of the confounder model for the provision of breast milk from the infants´ mother did not relevantly impact the *p*-value for the association of upper airway colonization with bacteria with pathogenic potential and the risk for moderate/severe BPD (*p* = 0.0025 versus *p* = 0.0043). As more than 90% of all children included in the study were fed with breast milk, there is too little variation in the covariate to see an effect in our analyses.

In summary, the colonization with bacterial strains with pathogenic potential during the first six weeks of life showed a statistically significant, yet unstructured, effect on the risk of moderate/severe BPD in a study population at high risk for functional limitations in lung function.

## 4. Discussion

The microbiome has a tremendous impact on long-term health. Alterations of microbiome structures in the preterm infant are coming more and more into the focus of research. While the beneficial effects of a healthy bacterial composition are well-known for the gut, the impact of the pulmonary microbiome on the pulmonary outcome still awaits a detailed investigation [10,15,16]. Our data add further evidence that specific alterations in bacterial colonization impact lung development and the severity of BPD in extremely low birth weight infants. While previous studies focused on subgroups of severely affected infants requiring mechanical ventilation and on colonization of the lower respiratory tract using tracheal aspirates [15,16], our data extend the knowledge to the whole cohort of infants at risk as continuously less infants require invasive mechanical ventilation due to the advances in non-invasive surfactant application and ventilation strategies. Microbial data were obtained during routine weekly screening and can therefore be easily adopted by the clinics to early on identify infants at special risk and to delineate therapeutic interventions with the aim to restore the physiologic situation of bacterial colonization. As the preterm infant is exposed to non-physiologic bacteria from the surrounding NICU [41,42] in a situation where the immune system is immature and incapable of bacterial clearance, its insufficient defense mechanisms might account for the highly significant result which is surprising taking into account the multifactorial origins of BPD [43,44].

It is well accepted that colonization of the lower respiratory tract with specific bacterial strains poses the preterm infant at increased risk for BPD [10,15,16,17]. Furthermore, the airway microbiome in infants developing BPD is different and less diverse than in infants not developing BPD and this distinction persists during the whole course of mechanical ventilation [45]. We provide novel data on the association between upper respiratory tract colonization and the degree of pulmonary disease. At first sight, our data are surprising as the absence of colonization with Gram-negative bacteria including potential pathogens as *E. coli* and *Klebsiella* species did not reduce the risk for BPD. On the other side, different bacterial strains display different inflammatory and immune system shaping properties. Gram-negative strains like *E. coli* and *Klebsiella* species are well known for their extensive activation of pro-inflammatory immune response during infection, but many other strains are also capable to do so including *Ureaplasma* species, which is frequently found in the newborn [46]. Therefore, the separation of bacteria with pathogenic potential and commensal strains might be superior to the differentiation by Gram staining and awaits further investigation in future studies.

Disease initiation and progression are broadly ascribed to the specific action of microbial axes especially from the gut including the gut–lung and gut–brain axes in many pediatric and adult diseases [47]. Here, our investigation revealed different effects of bacterial colonization of the upper respiratory tract and the gut that is in accordance with a recent publication that demonstrated a distinct colonization of the respiratory tract and gut in preterm infants [48]. It remains to be determined how antibiotic exposure shapes the bacterial composition of the upper airway. The tremendous impact of bacterial miscolonization on nosocomial infections, necrotizing enterocolitis, and survival rates has been long accepted and microbiota interventional studies were successful with respect to these study endpoints. In contrast, the approach did not reduce the incidence and severity of BPD [28,29]. This might be explained by the different microbial requirements of the immature lung for proper development and our results give a first hint into this direction. Furthermore, our data indicate that not only bacterial sepsis but pathogenic microbial colonization per se constitute a risk to the immature lung [26,27]. The number of cases with ROP ≥stage 3 or periventricular leukomalacia was too low to dissect statistically significant differences for further important morbidities of prematurity.

The complex signaling network in the lung comprises a tremendous overlap between inflammatory responses and lung growth promoting pathways. Abrogation of individual pathways revealed that the balanced equilibrium is a prerogative to preserve physiologic lung growth with NFκB taking center stage. Therefore, modulation of the inflammatory cytokine response and thereby NFκB in either direction puts the lung at risk for increased lung damage. BPD has been recently demonstrated to evolve from either inhibition or overstimulation of the pathways of classical pro-inflammatory cytokine response including TNF-α and IL-6 [49,50,51,52,53]. It is well accepted that pre- and postnatal bacterial infections aggravate lung injury via pro-inflammatory cytokine production [4,5,26,27,54]. The extent of NFκB activation can also be modified by specific pathogen colonization as observed in a gnotobiotic mouse model [55]. Consequently, our identified differences in upper airway bacterial composition could protect or disturb the pathway equilibrium and thus lung development. Due to the retrospective nature of our analyses, we could not associate bacterial colonization and the alteration in growth promoting and inflammatory properties. Future research is required to dissect this association.

Limitations of our study comprise the restriction to an association and the single center approach in a restricted number of cases. On the other hand, the distribution of bacterial strains within our cohort is diverse and within the expected range with dominance of strains like Coagulase-negative staphylococci, *E. coli*, *Enterobacterales*, and *Klebsiella* species in contrast to other studies where one dominating germ was identified as the causative factor [32,33,34,35]. Limitations are outweighed by the availability of datasets for detailed analyses (*n* = 100/102), the comprehensive risk adjustment for all relevant baseline confounders and the constantly small statistical *p*-values for the major outcome (*p* < 0.01). The validity of our data is increased by continuous protocols with regard to antibiotic therapy and ventilation strategies, not adding a further confounder to the study. Furthermore, BPD rates did not change significantly over time. Our data await confirmation in independent and larger cohorts before delineation of future prospective studies with the aim to modify bacterial colonization in the lung as is already well-established for the gut [28].

## 5. Conclusions

In summary, our data add evidence to another dimension of the multifactorial origins of BPD: the impact of bacterial colonization on the immature lung. It is not surprising that clinical routine measures like the provision of breast milk and the reduction of antibiotic exposure are able to reduce the burden of BPD considering the impact on microbiota structures [21,22,24,25,56]. Tremendous hopes for substantial improvements to prevent BPD arise from innovative approaches like recombinant cytokine application and mesenchymal stem cell therapy [57,58,59]. Targeting the microbiota of the upper airway might also be efficacious. Furthermore, our results clearly indicate that all clinical routine measures need to be reviewed in the context of microbiome modulation and with respect to the resulting consequences for the lung. Weekly routine bacterial screening of the upper airway using predefined categories seems a feasible approach to study the impact on the pulmonary outcome in all preterm infants independent of the ventilator status. Our screening approach can be easily extended to a plenty of scientific issues in neonatology and beyond.

## Figures and Tables

**Figure 1 jcm-09-02240-f001:**
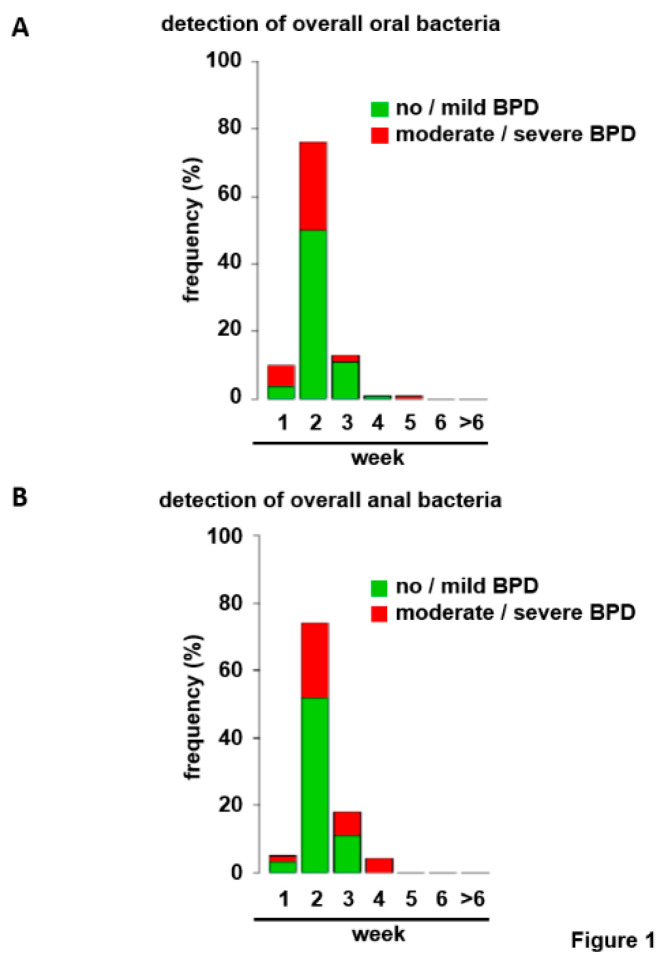
Frequency of overall bacterial colonization of the upper respiratory tract and the gut in the study cohort. Depicted are the frequencies of total bacterial colonization of the upper respiratory tract (**A**) and the gut (**B**) investigated by nasopharyngeal and anal swabs within the first six weeks of life. The additional category comprises the cases were no positive result was obtained within the observational period (>6). Green sections of bars represent infants with no/mild BPD, and red sections preterms with moderate/severe BPD.

**Figure 2 jcm-09-02240-f002:**
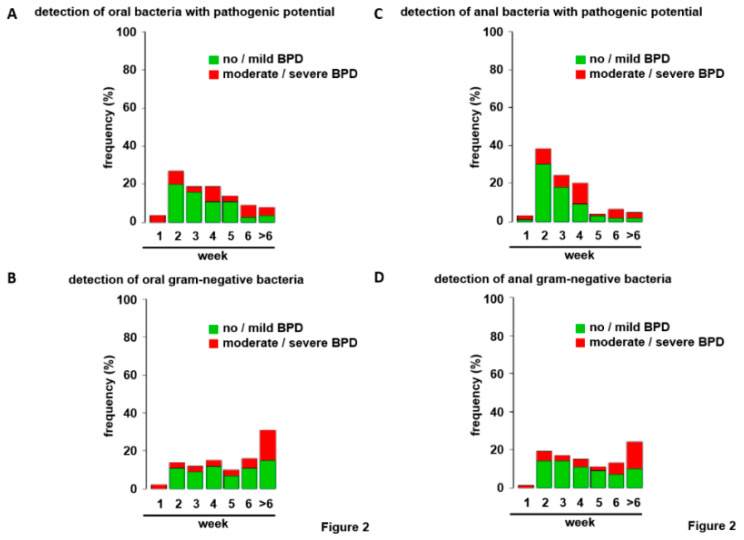
Frequency of colonization of the upper respiratory tract and the gut by bacterial categories within the study cohort. Depicted are the frequencies of bacterial colonization of the upper respiratory tract (**A**,**B**) and the gut (**C**,**D**) investigated by nasopharyngeal and anal swabs within the first six weeks of life. The additional category comprises the cases were no positive result was obtained within the observational period (>6). Bacterial strains were separated for their pathogenic potential (**A**,**C**) and Gram staining reaction (**B**,**D**). Graphics are displayed for BPD disease severity as in Figure 1.

**Table 1 jcm-09-02240-t001:** Bacteria with pathogenic potential separated by the gram staining response in alphabetical order.

Gram-Positive Strains	Gram-Negative Strains
Enterococcus species (*n* = 72)	Citrobacter species (*n* = 2)
Staphylococcus aureus (*n* = 44)	Enterobacter species (*n* = 28)
Streptococcus group A (*n* = 1)	Escherichia coli (*n* = 63)
Streptococcus group B (*n* = 2)	Haemophilus species (*n* = 2)
	Klebsiella species (*n* = 17)
	Proteus species (*n* = 4)
	Pseudomonas species (*n* = 6)
	Serratia species (*n* = 0)
	Stenotrophomonas maltophilia (*n* = 0)

Twelve species were classified as bacteria with pathogenic potential based on published literature and datasets from the German National Reference Center for the Surveillance of Nosocomial Infections [32,33,34,35], *n* indicates the number of preterm infants colonized at any detection site within the first six weeks of life.

**Table 2 jcm-09-02240-t002:** Patient cohort characteristics.

	Total Cohort	No or Mild	Moderate or Severe	*p*-Value
	(*n* = 102)	BPD (*n* = 67)	BPD (*n* = 35)	
gestational age (weeks)	26 + 3 (23 + 4 − 30 + 5)	26 + 6 (24 + 3 − 30 + 5)	25 + 4 (23 + 4 − 29 + 1)	*p* = 0.001 **
birth weight (g)	795 (320–995)	880 (490–995)	690 (320–990)	*p* < 0.001 **
male gender (%)	45/102 (44.1%)	24/67 (35.8%)	21/35 (60.0%)	*p* = 0.020 *
small for gestational age (%)	11/102 (10.8%)	2/67 (3.0%)	9/35 (25.7%)	*p* < 0.001 **
multiple birth (%)	47/102 (46.1%)	32/67 (47.8%)	15/35 (42.9%)	*p* = 0.660
bronchopulmonary dysplasia no (%)	18/102 (17.6%)	18/67 (26.9%)		-
mild (%)	49/102 (48.0%)	49/67 (73.1%)		
moderate (%)	5/102 (4.9%)		5/35 (14.3%)	
severe (%)	30/102 (29.4%)		30/35 (85.7%)	
intraventricular hemorrhage				
total incidence (%)	15/102 (14.7%)	9/67 (13.4%)	6/35 (17.1%)	*p* = 0.621
grade III or IV (%)	11/102 (10.8%)	7/67 (10.4%)	4/35 (11.4%)	*p* = 0.885
periventricular leukomalacia (%)	1/102 (1.0%)	1/67 (1.5%)	0/35 (0.0%)	*p* = 0.485
retinopathy of prematurity				
any stage (%)	66/102 (64.7%)	36/67 (53.7%)	30/35 (85.7%)	*p* = 0.001 **
≥stage 3 antenatal steroids	23/102 (22.5%)	7/67 (10.4%)	16/35 (45.7%)	*p* < 0.001 **
<24 h	14/102/13.7%)	8/67 (11.9%)	6/35 (17.1%)	*p* = 0.026 *
24 h—7 days	48/102 (47.1%)	27/67 (40.3%)	21/35 (60.0%)	
>7 days	40/102 (39.2%)	32/67 (47.58%)	8/35 (22.9%)	
provision of breast milk (%)	93/102 (91.2%)	62/67 (92.5%)	31/35 (88.6%)	*p* = 0.509
antibiotic therapy				
any (%)	90/102 (88.2%)	58/67 (86.6%)	32/35 (91.4%)	*p* = 0.475
started directly after birth (%)	76/102 (74.5%)	52/67 (77.6%)	24/35 (68.6%)	*p* = 0.325
duration antibiotic therapy (days)	5 (0–41)	5 (0–38)	9 (0–41)	*p* = 0.002 **

Patient characteristics are depicted for the total study cohort. Preterm infants were separated for the severity of bronchopulmonary dysplasia (BPD) according to the actual definition and grouped as infants with no or mild BPD and infants with moderate or severe BPD [29]. Gestational age is depicted in completed weeks of gestation plus additional days. The criterion small for gestational age was fulfilled with percentiles for body weight, length, and head circumference below the 10th percentile. Antenatal steroids were separated into provision <24 h, within 24 h and 7 days and beyond 7 days before delivery. Repeated application immediately before delivery was added to the category 24 h to 7 days. Provision of breast milk comprises all infants that received breast milk from its own mother within the first six weeks of life independent of the amount provided. Antibiotic therapy includes any application within the first six weeks of life and is presented for the variables any application, therapy initiated directly after birth and the total duration in total days. Statistical analysis was performed comparing infants with no or mild BPD and infants with moderate or severe BPD. * *p* < 0.05, ** *p* < 0.01.

**Table 3 jcm-09-02240-t003:** Confounder model to take additional risk factors for BPD into account.

	Estimate	Standard Error	z-Value	*p*-Value
(intercept)	8.8811	5.5682	1.595	0.1107
gestational age (days)	−0.0288	0.0331	−0.871	0.3838
birth weight (g)	−0.0078	0.0026	−3.023	0.0025 **
small for gestational age	2.1459	1.3547	1.584	0.1132
male gender	1.4725	0.6621	2.224	0.0262 *
multiple birth	0.4519	0.3816	1.184	0.2363
antenatal steroids	−0.0689	0.0396	−1.740	0.0818
(days before delivery)				
duration antibiotic therapy (days)	0.0660	0.0495	1.334	0.1822

For model selection, AIC-based, backward selection was employed, with risk factors gestational age and multiple birth added manually. Severity of bronchopulmonary dysplasia (BPD) was separated into no/mild BPD (0) and moderate/severe BPD (1) as the binary response. Covariates are gestational age, birth weight, small for gestational age (with 1: yes, 0: no), gender (1: male, 0: female), multiple birth (the number of multiples: 1, 2, 3, or 4), antenatal steroids (the number of days steroids were given before birth) and total duration of antibiotic therapy during the first six weeks of life (observation period). * *p* < 0.05, ** *p* < 0.01.

**Table 4 jcm-09-02240-t004:** Influence of the site and category of bacterial colonization within the first six weeks of life on BPD severity.

	Residual Degrees of Freedom	Residual Deviance	Degrees of Freedom	Deviance	*p*-Value
upper airway pathogenic bacteria	87	63.662	5	17.1102	0.0043 **
upper airway gram-negative bacteria	87	77.876	5	2.8959	0.7160
anal pathogenic bacteria	87	74.365	5	6.4067	0.2686
anal gram-negative bacteria	87	71.086	5	9.6856	0.0847

To check for statistically significant impact of germ localization (upper airway or anal) and germ category (pathogenic potential as defined in Table 1, or gram-negative bacterial strains), the confounder model was compared to the model with the respective germ included by analysis of deviance. For the item upper airway pathogenic bacteria, the effect was still significant on the 5% level after adjusting the *p*-value using Bonferroni correction. ** *p* < 0.01.

**Table 5 jcm-09-02240-t005:** Final model including information on upper airway pathogenic bacterial colonization and additional covariates.

Estimated	Regression Coefficient	Standard Error	z-Value	*p*-Value
(intercept)	14.7012	6.9322	2.121	0.0339 *
gestational age (days)	−0.0434	0.0395	−1.099	0.2716
birth weight (g)	−0.0119	0.0034	−3.552	0.0004 ***
small for gestational age	2.6637	1.4345	1.857	0.0633
gender	2.1162	0.7970	2.655	0.0079 **
multiple birth	0.8535	0.4926	1.733	0.0832
Antenatal steroids (days before delivery)	−0.0764	0.0474	−1.610	0.1073
duration antibiotic therapy (days)	0.0727	0.0709	1.025	0.3052
oral pathogenic bacteria/week 3	−3.6769	1.2262	−2.998	0.0027 **
oral pathogenic bacteria/week 4	−0.5549	0.9716	−0.571	0.5679
oral pathogenic bacteria/week 5	−2.3071	1.1739	−1.965	0.0494 *
oral pathogenic bacteria/week 6	−0.0651	1.1884	−0.055	0.9563
oral pathogenic bacteria/week >6	−0.1528	1.6931	−0.090	0.9281

Results of the logistic regression model. Severity of bronchopulmonary dysplasia (BPD) was separated into no/mild BPD (0) and moderate/severe BPD (1) as the binary response. The factor upper airway pathogenic bacteria is dummy-coded with “week ≤2” as the reference. Additional, potentially confounding covariates are gestational age, birth weight, small for gestational age (with 1: yes, 0: no), gender (1: male, 0: female), multiple birth (the number of multiples: 1, 2, 3, or 4), antenatal steroids (the number of days steroids were given before birth) and total duration of antibiotic therapy during the first six weeks of life (observation period). * *p* < 0.05, ** *p* < 0.01, *** *p* < 0.001.

**Table 6 jcm-09-02240-t006:** Alternative confounder model including antibiotic therapy started immediately after birth as covariate.

Estimate	Standard Error	z-Value	*p*-Value	Regression Cofficient
(intercept)	22.4478	7.8924	2.844	0.0045 **
gestational age (days)	−0.1007	0.0456	−2.208	0.0272 *
birth weight (g)	−0.0068	0.0027	−2.524	0.0116 *
small for gestational age	2.4235	1.4029	1.728	0.0841
male gender	1.3853	0.6866	2.018	0.0436 *
multiple birth	0.5009	0.3889	1.288	0.1977
antenatal steroids	−0.0667	0.0413	−1.617	0.1059
(days before delivery)				
antibiotic therapy started immediately after birth	−1.6629	0.8801	−1.889	0.0588

The model is analog to Table 3, but antibiotic exposure is included differently. Severity of bronchopulmonary dysplasia (BPD) was separated into no/mild BPD (0) and moderate/severe BPD (1) as the binary response. Covariates are gestational age, birth weight, small for gestational age (with 1: yes, 0: no), gender (1: male, 0: female), multiple birth (the number of multiples: 1, 2, 3, or 4), antenatal steroids (the number of days steroids were given before birth) and antibiotic therapy started immediately after birth (yes/no). * *p* < 0.05, ** *p* < 0.01.

**Table 7 jcm-09-02240-t007:** Altered confounder model including any antibiotic therapy during the first six weeks of life as covariate.

Estimated	Standard Error	z-Value	*p*-Value	Regression Coefficient
(intercept)	10.5035	5.8628	1.792	0.0732
gestational age (days)	−0.0361	0.0338	−1.071	0.2841
birth weight (g)	−0.0085	0.0027	−3.110	0.0019 **
small for gestational age	2.1268	1.3412	1.586	0.0887
male gender	1.7519	0.6869	2.550	0.0108 *
multiple birth	0.4434	0.3746	1.184	0.2366
antenatal steroids	−0.0698	0.0400	−1.746	0.0807
(days before delivery)				
antibiotic therapy (yes/no) within the first 6 weeks of life	0.6543	1.1178	0.585	0.5583

The model is analog to Table 3, but antibiotic exposure is included as a binary factor. Severity of bronchopulmonary dysplasia (BPD) was separated into no/mild BPD (0) and moderate/severe BPD (1) as the binary response. Covariates are gestational age, birth weight, small for gestational age (with 1: yes, 0: no), gender (1: male, 0: female), multiple birth (the number of multiples: 1, 2, 3, or 4), antenatal steroids (the number of days steroids were given before birth) and any antibiotic therapy during the first six weeks of life (observation period) independent of the duration of application. * *p* < 0.05, ** *p* < 0.01.

**Table 8 jcm-09-02240-t008:** Impact of different definitions of the covariate antibiotic exposure on the results for oral pathogenic bacteria and BPD.

	Residual Degrees of Freedom	Residual Deviance	Degrees of Freedom	Deviance	*p*-Value
duration antibiotic therapy (days)	87	63.662	5	17.110	0.0043 **
antibiotic therapy started	87	61.739	5	17.199	0.0041 **
immediately after birth					
antibiotic therapy (yes/no)	87	64.477	5	17.912	0.0031 **
within the first 6 weeks of life					

To check for robustness of the results for upper airway bacterial localization and the category pathogenic potential as defined in Table 1 against different definitions of antibiotic exposure, analysis of deviance as done in Table 4 was repeated for varying definitions of antibiotic exposure in the confounder model. For all definitions of antibiotic exposure applied, the effect was still significant on the 5% level after adjusting the *p*-value using Bonferroni correction. ** *p* < 0.01.

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
