# Peer review of "Bacterial Colonization within the First Six Weeks of Life and Pulmonary Outcome in Preterm Infants <1000 g"

_jcm, 2020, doi:10.3390/jcm9072240_

Round 1

Reviewer 1 Report

The article is very interesting that emphasizes the role of bacterial colonization in the development of a type of CLD. In addition, bacterial colonization is different depending on the digestive tract being analyzed. It is important to explore the results of the microbiota in all adverse neonatal outcomes since infections, NEC or even ROP are morbidities that develop during the neonatal period, especially in the preterm infants, and that connection throughout life. However, I believe that the results of the work need to be extensively reviewed. My doubts are that both the models and the adjustment variables to them do not support the conclusions drawn. I would like to comment on some of the recommendations to the authors, whom I thank for their effort in reporting these data.

Introduction: in general, is a well described section, however the authors need to pay attention to some sentences which need to be supported by references. In the other hand, the preterm infant is a key concept in the paper, which is not defined. This is important because not all preterm infant is the same and the best approach is the gestational age and the implication of the develop of lung around 32 week of gestation.

  • Lines 51-52, some references to support this observation would be interesting. In addition, the decrease in bacterial biodiversity in utero would be a risk factor to preterm labor? The line 56 derived from some research article which should be cited. The sentence reported in line 63 also should be cited.
  • Why birth weight <1000g at special risk for BPD? Some details need to be reported in the section. In addition, why the first 6 weeks of life constitutes vulnerable period? the majority of the diagnoses in neonatal outcomes happens around first 14 days of life, however, the greater prematurity, the longest time to diagnose neonatal adverse outcomes.

Experimental section: In general is a focus section, however the Data acquisition and parameter definition section should be reviewed the particularly clinical definition to increase the compressibility (i.e., the states of BPD or the SGA according to echography).

  • Line 92, maybe the authors could explain more the inclusion and exclusion criteria, which are follow-up described, instead of report the reference 23 which is unclear. Furthermore, the exclusion criteria and the sample removed should be re-wrote to increase the understanding.
  • Additionally, information related to nutritional intake could be interesting, the infants were fed own breast milk or donor breast milk, some infant was supplemented of fortifiers their feedings? At what day the parenteral route was extracted, when the gold enteral feeding was getting?
  • In the statistical analysis, the “Whitney rank sum” test is known as “Mann Whitney U test” or “Wilcoxon Rank Sum Test”. Not always a p<0.05 is usual. In the other hand, What packages were used in R? Was interfaces of RStudio applied?

Results: I do not believe the analysis is reported in the right direction. First of all, in the bacterial colonization of the upper respiratory tract and the gut, some test would be reported the different between colonization place. Secondly, the regression models did not include the any strains in the bacterial diversity or any score which authors calculated (i.e. table 3 and 4 need to be combined the predictor variables). In addition, why is necessary the intercept in the models if the authors did not build the equations? and more, the models would be more readable if the authors reported the OR and 95% of confidence intervals. According to the sentence: “In summary, the colonization with bacterial strains with pathogenic potential during the first 6 weeks of life showed a statistically significant, yet unstructured, effect on the risk of moderate/severe BPD in a study population at high risk for functional limitations in lung function.” I cannot found the results to support that.

  • I suggest that the sentence “To dissect the impact of specific patterns of microbial colonization and BPD we focused our analysis on preterm infants with a birth weight <1000 g and below 32 weeks of gestation that have the highest risks for relevant limitations in lung function (n=102)” move in the previous section, because synthetize the cohort and also the study design.
  • The footnote of the table 2 is cut. In addition, I suggest the authors need to clarify what group compere to what and obtain the p-value.
  • Some part of the section would be discussion section.

Discussion:

  • The sentence in the line 287 need a cite to support it.
  • The sentence “As the preterm infant is massively exposed to non-physiologic bacteria from the surrounding NICU” could be uncertainly, because most of the NICUs are high level of assistance which included restrictive politics to visit, in addition the technology to care the preterm infants mostly are very restrictive.
  • In other hands, the authors mentioned the Klebsiella spp. which could be the most important strain to develop not only sepsis (LOS) but also NEC in preterm infants. Some preventive antibi-therapy could be decrease this strains, among others, to control the LOS and NEC. What the authors opinion about the control their results by drugs treatment and others comorbidity? At the same time to prevent morbidities, we would be preventing BPD severity?
  • Additionally, the oxygen treatment, which prevent BPD, is a risk factor to develop ROP. What was this point in the results context of this manuscript?

Minor comments:

  • Line 50: Reference would be [6-9]
  • Line 59: references also would be [10-12]
  • Line 72: the concept of “late onset infections” is similar to “Late onset sepsis”?
  • Line 111 and 123: Need to be the country.
  • Line 117: references would be [23-26]
  • Line 283: references would be [9-11]
  • The table would be placed as an editable-table

Author Response

Comment: The article is very interesting that emphasizes the role of bacterial colonization in the development of a type of CLD. In addition, bacterial colonization is different depending on the digestive tract being analyzed. It is important to explore the results of the microbiota in all adverse neonatal outcomes since infections, NEC or even ROP are morbidities that develop during the neonatal period, especially in the preterm infants, and that connection throughout life. However, I believe that the results of the work need to be extensively reviewed. My doubts are that both the models and the adjustment variables to them do not support the conclusions drawn. I would like to comment on some of the recommendations to the authors, whom I thank for their effort in reporting these data.

Response: We thank the reviewer for having taken the time to thoroughly review our manuscript and we highly appreciate the overall positive estimate and the valuable comments to improve our manuscript. Within specific point 7 and 8 we respond to the statistical questions raised. Where appropriate we included additional references to fundament our statements.

1.) Introduction:

  • in general, is a well described section, however the authors need to pay attention to some sentences which need to be supported by references. In the other hand, the preterm infant is a key concept in the paper, which is not defined. This is important because not all preterm infant is the same and the best approach is the gestational age and the implication of the develop of lung around 32 week of gestation.

Response: As suggested we specified inclusion criteria and the special high vulnerability of infants with a birth weight <1000g within the cohort of infants ≤ 32 weeks of gestation.

Changes to the manuscript: Lane 82-85: “…to identify preterm infants with a birth weight <1000g and gestational age ≤ 32+0 weeks at special risk for BPD. The inclusion criteria were selected based on the particularly high vulnerability for BPD in this population of infants with lungs in the saccular stage of lung development [30,31].”

  • Lines 51-54, some references to support this observation would be interesting. In addition, the decrease in bacterial biodiversity in utero would be a risk factor to preterm labor? The line 56 derived from some research article which should be cited. The sentence reported in line 61-64 also should be cited.

Response and changes to the manuscript: As suggested by the reviewer, we included references 11,12,13,4 and 19,20 at the corresponding text passages.

  • Why birth weight <1000g at special risk for BPD? Some details need to be reported in the section. In addition, why the first 6 weeks of life constitutes vulnerable period? the majority of the diagnoses in neonatal outcomes happens around first 14 days of life, however, the greater prematurity, the longest time to diagnose neonatal adverse outcomes.

Response: The decision to include infants with a birth weight <1000g is explained under            specific point 1. According to the time course of bacterial colonization (please refer to Figure 1 and 2) and statistical results (lane 283-285). Statistical analysis was repeated after excluding infants from the category “no detection of bacteria until week 6”. However, results did not change substantially (p=0.0038).) the first 6 weeks of life prevailed as the decisive period.

Changes to the manuscript: Lane 88-89: “…within the first 6 weeks of life which constitute the decisive period from birth where most infants were colonized.”

Lane 220: “After the first 6 weeks of life, most infants were colonized (Figure 2A-D).”

2.) Experimental section:

  • In general is a focus section, however the Data acquisition and parameter definition section should be reviewed the particularly clinical definition to increase the compressibility (i.e., the states of BPD or the SGA according to echography).

Response: As suggested, we highlighted the definitions used for BPD stages and SGA status applied.

Changes to the manuscript: Lane 145-149: “Small for gestational age (SGA) status was defined as all three parameters of birth weight, length and head circumference below the 10th percentile according to the percentiles from the German perinatal registry [28]. Severity of BPD was separated according to the current NIH consensus definition as no, mild, moderate or severe BPD [2].”

  • Line 94, maybe the authors could explain more the inclusion and exclusion criteria, which are follow-up described, instead of report the reference 23 which is unclear. Furthermore, the exclusion criteria and the sample removed should be re-wrote to increase the understanding.

Response and changes to the manuscript: We removed former reference 23. The exclusion criteria and numbers of patients excluded are described in detail in the subsequent sentence (lane 94-101: “n=37 preterm infants were excluded due to death before 36 weeks of gestation (n=28), severe congenital malformations of the heart, brain, gut or urogenital tract (n=7) or transfer to another center before 36 weeks of gestation (n=2). Furthermore, all infants with severe gastrointestinal complications (n=7) of necrotizing enterocolitis or focal intestinal perforation were excluded…)”.

  • Additionally, information related to nutritional intake could be interesting, the infants were fed own breast milk or donor breast milk, some infant was supplemented of fortifiers their feedings? At what day the parenteral route was extracted, when the gold enteral feeding was getting?

Response: We thank the reviewer for the valuable comment. We included a detailed description of breast milk supply, milk fortification, nutrition therapy based on the ESPGHAN recommendations and the criteria for discontinuation of parenteral nutrition.

Changes to the manuscript: Lane 152-156:” Provision of breast milk from the baby´s mother was supported in the absence of a milk donor program and breast milk was fortified with Aptamil FMS 4,4% (Milupa, Frankfurt am Main, Germany). Nutritional supply was provided according to the actual recommendations and parenteral nutrition was discontinued when nutritional requirements were met by the enteral supply [39].”

  • In the statistical analysis, the “Whitney rank sum” test is known as “Mann Whitney U test” or “Wilcoxon Rank Sum Test”. Not always a p<0.05 is usual. In the other hand, What packages were used in R? Was interfaces of RStudio applied?

Response: Thanks for pointing this out. No additional packages were needed for the analyses performed in R. The base distribution is sufficient.

Changes to the manuscript: Lane 162: Whitney rank sum test” has been replaced by “Mann-Whitney U test”; Lane 166: the information on R (R Base Distribution) has been added.

3.) Results:

  • I do not believe the analysis is reported in the right direction. First of all, in the bacterial colonization of the upper respiratory tract and the gut, some test would be reported the different between colonization place. Secondly, the regression models did not include the any strains in the bacterial diversity or any score which authors calculated (i.e. table 3 and 4 need to be combined the predictor variables). In addition, why is necessary the intercept in the models if the authors did not build the equations? and more, the models would be more readable if the authors reported the OR and 95% of confidence intervals. According to the sentence: “In summary, the colonization with bacterial strains with pathogenic potential during the first 6 weeks of life showed a statistically significant, yet unstructured, effect on the risk of moderate/severe BPD in a study population at high risk for functional limitations in lung function.” I cannot found the results to support that.

Response: You are right, the information on bacterial colonization and the additional, potentially confounding covariates need to be considered jointly. And this is exactly what has been done (compare the new Table 5 below). The corresponding logistic regression model is then compared to the confounder model (Table 3, where the information on bacteria is removed) to check for significant overall effects of the corresponding factors (Table 4). The intercept is an essential part of the model, among other things, because we used dummy coding for the factors describing bacteria colonization. For supporting our conclusions, we also included the complete information on the final model regressing BPD on oral pathogenic colonization and the additional, potentially confounding covariates (see also below). Since units are very different between covariates (days, g, dummies, etc.), we did not give exp(b) (i.e., OR) as this may mislead interpretation. Instead of confidence intervals, we give standard errors (as 95% confidence intervals can be computed easily as +/- 2 se).

Changes to the manuscript: As a new Table 5, the complete results of modeling the influence of upper airway pathogenic bacteria colonization have been added, and results are discussed in Section 3.2 (lane 285-300). 

  • I suggest that the sentence “To dissect the impact of specific patterns of microbial colonization and BPD we focused our analysis on preterm infants with a birth weight <1000 g and below 32 weeks of gestation that have the highest risks for relevant limitations in lung function (n=102)”move in the previous section, because synthetize the cohort and also the study design.

Response and changes to the manuscript: As suggested by the reviewer, we moved this sentence into the previous section (now lane 159-161).

  • The footnote of the table 2 is cut. In addition, I suggest the authors need to clarify what group compere to what and obtain the p-value.

Response and Changes to the manuscript: We thank the reviewer for the hint and now present the total footnote. The comparison of infants with no/mild versus moderate/severe BPD is now annotated in the legend. Absolute p-values are presented in the last column of the table.

  • Some part of the section would be discussion section.

Response: We included detailed descriptions and citations of the underlying literature to ease the understanding of the experimental approach for the reader. We would be happy to move the intended parts into the discussion section if the reviewer specifies them.

4.) Discussion:

  • The sentence in the line 287 need a cite to support it.

Response and changes to the manuscript: We moved the citations at the end of the section referring to the mentioned studies upwards. Lane 379-381: “While previous studies focused on subgroups of severely affected infants requiring mechanical ventilation and on colonization of the lower respiratory tract using tracheal aspirates [15,16],…”.

  • The sentence “As the preterm infant is massively exposed to non-physiologic bacteria from the surrounding NICU” could be uncertainly, because most of the NICUs are high level of assistance which included restrictive politics to visit, in addition the technology to care the preterm infants mostly are very restrictive.

Response and changes to the manuscript: We thank the reviewer for his valuable comment. We removed “massive” and included references 41,42 to fundament our statement (lane 386-387).

  • In other hands, the authors mentioned the Klebsiella spp. which could be the most important strain to develop not only sepsis (LOS) but also NEC in preterm infants. Some preventive antibi-therapy could be decrease this strains, among others, to control the LOS and NEC. What the authors opinion about the control their results by drugs treatment and others comorbidity? At the same time to prevent morbidities, we would be preventing BPD severity?

Response: We extensively monitored clinically important categories of antibiotic therapy as presented in line 323-326 where we did not find any positive effect of antibiotic exposure: “To further dissect the impact of antibiotic exposure on the colonization of the upper airway with bacteria with pathogenic potential in the study cohort, the total duration of antibiotic therapy measured in days was replaced by the covariates “antibiotic therapy started immediately after birth” or “any antibiotic therapy within the first 6 weeks of life”.” We introduce the pioneering results on the negative consequences of antibiotic exposure for the pulmonary outcome within lane 71-72: “Furthermore, recent studies demonstrated that antibiotic therapy per se and its prolonged use was associated with an increased risk for BPD [23,24,25].” Based on previous and our own results we cannot derive a benefit of prophylactic antibiotic therapy.

  • Additionally, the oxygen treatment, which prevent BPD, is a risk factor to develop ROP. What was this point in the results context of this manuscript?

Response: The number of subjects included into our analysis was scheduled for the outcome BPD based on the published incidence (reference 31).

Changes to the manuscript: Lane 416-418: “The number of cases with ROP ≥ stage 3 or periventricular leukomalacia was too low to dissect statistically significant differences for further important morbidities of prematurity.”

Minor comments:

  • Line 50: Reference would be [6-9]

Response and changes to the manuscript: We thank the reviewer for critically revising our manuscript and detecting the numbering error. We rechecked and corrected all references.

  • Line 59: references also would be [10-12]

Response and changes to the manuscript: We thank the reviewer for critically revising our manuscript and detecting the numbering error. We rechecked and corrected all references.

  • Line 72: the concept of “late onset infections” is similar to “Late onset sepsis”?

Response and changes to the manuscript: We thank the reviewer for his advice and exchanged the term “infection” by “sepsis” (lane 72).

  • Line 112 and 139: Need to be the country.

Response and changes to the manuscript: At the requested sites, we added the country (USA).

  • Line 117: references would be [23-26]

Response and changes to the manuscript: We thank the reviewer for critically revising our manuscript and detecting the numbering error. We rechecked and corrected all references.

  • Line 283: references would be [9-11]

Response and changes to the manuscript: We thank the reviewer for critically revising our manuscript and detecting the numbering error. We rechecked and corrected all references.

  • The table would be placed as an editable-table

Response and changes to the manuscript: As suggested, we changed the table into an editable table.

Reviewer 2 Report

I have reviewed the manuscript entitled ” Bacterial colonization within the first 6 weeks of life 2 and pulmonary outcome in preterm infants <1000g” The study is a single centra study from Germany analysing the impact of bacterial colonization of the upper airway and gastrointestinal tract on BPD development.I find the manuscript interesting and well written, and think it minor corrections.

General comment

I find it interesting and well done that the group have used cultures and not 16s as the technique to study bacterial colonization of the upper airway and gastrointestinal tract.

Specific comments

2.2 line 117: Coagulase-negative staphylococci were not classified as pathogenic due to their ubiquity in preterm infants despite the relevant contribution to catheter-associated infections.

I do not agree with this statement. I pretem infants CoNS should be considered as potentially pathogenic. CoNS are often the cause of sepsis in pretem infants

Table 1: Group A and B streptococci are put together in one group (n=3) Where there really any cases with group A streptococci, or were they all Group B?

Table 2: Does antenatal steroids < 24 h include the group of patients that did not receive antenatal steroids at al?

In general I think the study is interesting and well written and can be published after minor corrections.

Author Response

Comment: I have reviewed the manuscript entitled ” Bacterial colonization within the first 6 weeks of life 2 and pulmonary outcome in preterm infants <1000g” The study is a single centra study from Germany analysing the impact of bacterial colonization of the upper airway and gastrointestinal tract on BPD development.I find the manuscript interesting and well written, and think it minor corrections.

General comment

  • I find it interesting and well done that the group have used cultures and not 16s as the technique to study bacterial colonization of the upper airway and gastrointestinal tract.

Response: We want to express our deep thanks for having found the time to review our manuscript and for the very positive assessment of our study. The reviewer highlights one important point of our study that we rely on cultures that can be easily obtained and should be considered as additional technique or alternative in future studies.

Specific comments

  • line 118: Coagulase-negative staphylococci were not classified as pathogenic due to their ubiquity in preterm infants despite the relevant contribution to catheter-associated infections.

I do not agree with this statement. I pretem infants CoNS should be considered as potentially pathogenic. CoNS are often the cause of sepsis in pretem infants

Response: We agree with the reviewer that CoNS pose a potential threat to the preterm infant. As most infants in our cohort were colonized with CoNS within the first 6 weeks of life, a statistical analysis was impossible.

Changes to the manuscript: Lane 118-120: “Due to the high prevalence of colonization with coagulase-negative staphylococci in 95 out of 102 infants (93,1%), a statistical analysis to dissect its impact on BPD was not applicable to our cohort”.

  • Table 1: Group A and B streptococci are put together in one group (n=3) Where there really any cases with group A streptococci, or were they all Group B?

Response: There was n=1 case with streptococci Group A and n=2 with streptococci Group B.

Changes to the manuscript: For the clearness of presentation we separated streptococci Group A and B in Table 1.

  • Table 2: Does antenatal steroids < 24 h include the group of patients that did not receive antenatal steroids at al?

Response: We clarified the categorization of antenatal steroids < 24 h in the experimental section. Four mother-infant dyads did not receive antenatal steroids before birth.

Changes to the manuscript: Lane 143-144: “... including four cases with no antenatal steroid application before birth…”.

  • In general I think the study is interesting and well written and can be published after minor corrections.

Response: We thank the reviewer very much for the recommendation.

Round 2

Reviewer 1 Report

I would like to thank the authors for their efforts in re-editing the article. It has been considerably improved. No consideration on my part.